# Effect of Nanobubbles on Ultrafiltration Membrane Performance and Properties of Model Cheese Whey

**DOI:** 10.3390/foods14244224

**Published:** 2025-12-09

**Authors:** FNU Akshit, Sanjay Kumar Bharti, Maneesha S. Mohan

**Affiliations:** Alfred Dairy Science Laboratory, Dairy and Food Science Department, South Dakota State University, Brookings, SD 57007, USA; akshitbastali@gmail.com (F.A.); drskbharti@gmail.com (S.K.B.)

**Keywords:** flux, milk proteins, ultrafiltration, fouling, nanobubbles

## Abstract

Fouling has been a major concern in membrane processing, requiring frequent cleaning, increasing the time of processing, and reducing the lifespan of membranes. As a strategy to improve membrane filtration processes, our study investigates the impact of nanobubbles (NBs) on the whey ultrafiltration (UF) process and provides insights into the resulting changes in permeation flux, concentration factor, composition, particle charge and size, viscosity, and protein secondary structures. NBs led to significantly enhanced permeation flux up to 60 min (*p* < 0.05), leading to a higher concentration factor with time, as indicated at 120 min compared to the control. There was a significant increase in protein and total solids concentrations (33 ± 10% and 28 ± 5%) in the final retentate at 120 min for NB-treated cheese whey (NBW) as compared to the control (*p* < 0.05). While particle size is relatively unchanged with and without NB treatment, increased viscosity in NBW is caused by the increased concentration factors achieved with higher flux for the NBW by the end of UF. FTIR and SDS-PAGE reveal no significant alterations in whey protein secondary structures and fractions, respectively. Overall, membrane efficiency was enhanced by significantly increasing peak flux and concentration factor (34 ± 5% and 40 ± 4%) for NBW compared to CW. Our study presents an innovative approach to reach the targeted total solids/protein for cheese whey concentration in significantly less processing time (28.27% ± 2.33 reduction), with potential energy savings. Therefore, nanobubble technology shows promising potential to improve membrane filtration in the dairy industry with higher permeation flux, reduced fouling, and improved membrane processing efficiency.

## 1. Introduction

The dairy industry’s interest in membrane processing is primarily driven by its nonthermal nature, which enables the conservation of the native state and functional properties of milk components, especially milk proteins [1,2]. Among all membrane processes, ultrafiltration (UF) stands out as the most popular in the dairy industry, particularly for concentrating milk proteins from various streams such as skim milk and whey [1]. Whey is a by-product of the dairy industry, produced during the manufacturing of cheese, yoghurt, and caseinates. The general composition of whey is 93–94% water, 4.5–5% lactose, 0.7–0.9% ash, and 1–2% milk fat [2,3]. UF membranes with a 10 kDa molecular weight cut-off are commonly employed in the dairy industry. Milk proteins, being larger than 10 kDa, are concentrated in the retentate, and approximately 75% of UF membranes in the dairy industry are utilized for the fractionation/concentration of whey proteins [4].

Despite the widespread use of UF in the dairy industry, it is associated with challenges such as a reduction in permeation flux over time due to increased membrane fouling and/or concentration polarization layer [2,3]. Fouling refers to the deposition or aggregation of milk solids (Figure 1), particularly proteins and minerals, on the membrane’s surface and within its pores, while concentration polarization is the accumulation of milk solids over the surface of the membrane [5,6]. During the processing of whey, major whey proteins involved in fouling are α-lactalbumin, bovine serum albumin (BSA), and β-lactoglobulin [3]. Despite various alternative developments such as gas sparging [7], ultrasonication [2,8], higher pressures and cross-flow velocities, microturbulence [9], and vibration/rotating disc modules [10], fouling remains a major concern limiting permeation flux during UF processes. Moreover, membrane fouling increases plant maintenance and operating costs due to frequent cleaning cycles, shortened run times, and reduced membrane lifespan [9], necessitating pre-treatment and/or novel scalable alternatives to mitigate/reduce fouling during UF processes.

Recently, nanobubbles (NBs) have piqued the attention of researchers and have been proven instrumental in improving membrane performance by reducing fouling and increasing permeation flux. NBs are gaseous entities with spherical shapes ranging from 1 to 1000 nm, generated in the bulk liquid using various techniques such as hydrodynamic cavitation, acoustic cavitation, and membrane-based methods [6]. Earlier, Wu et al. [11] demonstrated that NB could alleviate membrane fouling for BSA, while Ghadimkhani et al. [12] showed that air NBs have the ability to clean fouled ceramic membranes without using any chemicals for cleaning. Recently, NBs enhanced permeation flux during the UF of humic acid, BSA, and sodium alginate solutions for drinking water applications [13]. Similarly, another study illustrated the ability of NBs to enhance permeate flux and reduce fouling for skim milk dispersions in constant concentration mode at 25% total solids [14]. Nanobubbles enhance membrane performance by increasing the permeation flux and delaying the formation of gel layer and fouling, attributed to shear forces created by bubble implosion at the membrane surface [12,14]. However, there is a gap in the literature, especially for dairy systems such as cheese whey, with regard to understanding the effect of NBs during ultrafiltration on membrane efficiency, simulating real-time industry conditions, and also evaluating the impact on whey proteins’ functional properties that could affect downstream processing. Therefore, the objective of this study was to evaluate the impact of NBs generated in model cheese whey on permeation flux and concentration factor during a lab-scale UF process. We hypothesize that the generation of nanobubbles in cheese whey could significantly improve efficiency during ultrafiltration by enhancing the permeation flux and reducing processing time.

## 2. Materials and Methods

### 2.1. Materials

Whey protein concentrate (WPC) powder (ISO Chill 3400 HF) was obtained from Agropur, Appleton, WI, USA. The WPC had a protein content of 35.3% and 8% fat on a dry basis, as specified in the manufacturer’s data sheet. Lactose (80–100 mesh) with a purity of 99.5% was procured from Milk Specialties Global, Eden Prairie, MN, USA. All chemicals used in the experiments were sourced from Fisher Scientific (Saint Louis, MO, USA) or Millipore Sigma (Burlington, MA, USA).

### 2.2. Formulation of the Model Whey

A model whey stream was prepared by mixing ISO Chill 3400 whey protein concentrate (WPC) with lactose added in calculated amounts in water to replicate the composition of actual whey. Specifically, 115 g of WPC and 154 g of lactose were dissolved in 4500 mL of water. The mixture underwent stirring for one hour using a magnetic stirrer at 400 RPM, followed by overnight hydration at 4 °C. The pH of the model cheese whey after overnight hydration was 6.86 ± 0.01. Three separate batches of cheese whey were manufactured on two different days for the three replicates and two treatments (Control (CW) and nanobubble-treated (NBW)), respectively.

### 2.3. Air Nanobubble Generation

Air nanobubbles were generated using a high-pressure swirling flow hydrodynamic cavitation design-based equipment called an oxydoser (Wicked Hydro Deals, Littleton, CO, USA). The oxydoser was immersed a few inches below the surface of the cheese whey collected in a 5-gallon bucket, and the inlet of the oxydoser was connected to the outlet of a centrifugal pump (Lawn transfer booster pump, Utilitech #148008, Lowes, SD, USA), pumping with a fixed flow rate at 49.2 L per minute (LPM). To continuously generate nanobubbles, recirculation was performed for 10 min by connecting the pump inlet to the outlet valve at the bottom of the bucket. Compressed air (AO-X Welding Supply, Sioux Falls, SD, USA) from a gas cylinder was utilized as the gas source for the Oxydoser, at a fixed 20 PSI pressure and 0.25 LPM air flow rate. Oxydoser had the outlets at the bottom and sides, and continuously generated nanobubbles in the bulk liquid while recirculating for 10 min, when the gas is connected. All the nanobubble generation experiments were conducted at room temperature and were performed using the optimized process setting from our previous studies [15]. 

### 2.4. Ultrafiltration of Whey

Following air nanobubble generation in the model cheese whey, ultrafiltration (UF) was conducted using the benchtop plate and frame Optisep 400 UF filtration module (Smart Flow Technologies, Sanford, NC, USA). The module was equipped with two flat sheets of 10 kDa polyether sulfone membrane (Synder Filtration, Vacaville, CA, USA). A masterflex L/S peristaltic pump (Cole Parmer, Thermo Fischer Scientific, Waltham, MA, USA) was used to supply feed to the UF membrane module, and a constant pressure of 50 PSI was maintained throughout the run. The constant pressure was maintained by using a back-pressure valve at the retentate outlet. Each run was conducted using 3.5 L of whey, with a fixed runtime of 120 min, while the concentration factor was calculated based on volume reduction in 120 min of run time for control and treated cheese whey. The retentate was recirculated to the feed container, while the permeate was collected in a conical flask, being constantly weighed on an electronic balance to record the permeate weight. Measurements for permeate weight were taken every 20 min to calculate the flux, and samples were also collected in 50 mL centrifuge tubes at the same time points for further analysis. Between trials, the UF system was rinsed with potable water, followed by cleaning using a warm soap solution, and then again, a final rinse using potable water. The membrane sheets were replaced with new 10 kDa membranes for each trial conducted. Initial membrane performance was calibrated by measuring water flux at transmembrane pressures of 50 PSI to establish baseline permeation flux.

### 2.5. Membrane Permeation Flux and Concentration Factor

The permeation flux rate was calculated based on the mass of the permeate collected at intervals of 20 min during the 120 min total run time of UF. Flux was calculated as kg/h m^2^, using the ratio of permeate flow rate (kg/h) to the total membrane surface area (0.04 m^2^). The calculation of permeate flux at regular intervals determined the effects of nanobubbles on membrane performance and fouling alleviation as compared to the control. The percentage increase in peak permeation flux was calculated for the point of highest permeation flux during the UF time by using the formula below:% Increase in Peak Flux = ((Peak flux for NBW − Peak flux for CW))/(Peak flux for CW) × 100

Peak permeation flux refers to the highest flux value throughout the total 120 min run time for UF for control and nanobubble-treated cheese whey.

Concentration factor (CF) was monitored by recording the ratio of weight reduction in retentate at each time point to the initial weight of feed used in the experiments. The percentage increase in concentration factor was calculated for the final retentate at 120 min using the formula below:% Increase in CF = (CF for NBW retentate − CF for CW retentate)/(CF for CW retentate) × 100

### 2.6. Total Solids, Protein, and Ash Content

Total solids (TS) and ash content were determined using standard gravimetric methods. The total solids content was determined using forced-air oven drying at 100 °C for 5–6 h, while ash was determined by placing the samples in a muffle furnace at 550 °C after evaporating residual moisture over a hot plate. Total protein content was analyzed using the Kjeldahl method [16]. Duplicate measurements were conducted for total solids, protein, and ash content in three trials.

The percentage increase in total solids and protein content was calculated for the final retentate at 120 min using the formula below:% Increase in TS/Protein = ((TS/protein for NBW − TS/protein for CW))/(TS/protein for CW) × 100

### 2.7. Particle Size and Zeta Potential

Particle size and zeta potential were measured by using Litesizer 500 from Anton Paar, Ashland, VA, USA. Particle size and zeta potential were measured by diluting the samples 1:100 to achieve a transmittance above 85%. The sample was then placed into the omega cuvettes supplied (Anton Paar, Ashland, VA, USA), and these were then placed in the sample holder. The instrument software (Kalliope, Anton Paar, Ashland, VA, USA) gave the particle size distribution data for the light scattering intensity of particles using the Einstein–Stokes equation. Then, the particle size mode values were calculated based on the light scattering intensity of the particle by plotting the particle size distribution data using Origin Pro software (Origin 2024b 10.15, OriginLab Corporation, Northampton, MA, USA).

Zeta potential (mV) was measured based on the electrophoretic mobility of the system by applying a constant voltage (200 V). The model of approximation used was Smoluchowski, with 1.50 as the Henry factor. The mean values obtained from the Kaliope software v4.13.3 for zeta potential have been reported with the standard error of the mean.

### 2.8. Viscosity

The viscosity of the control and NB-treated whey was measured using a stress–strain-controlled rheometer (MCR92, Anton Par, Graz, Austria). The analysis was performed using a concentric cylinder (CC39, 38.69 mm) and cup (C-CC39, 42.01 mm) measuring system at 20 °C. Approximately 30 mL of the sample was filled into the measuring cup, and the bob was dipped into the liquid, followed by starting the measurement. The sample was equilibrated at 20 °C for 1 min, and flow curves were obtained at shear rates between 0.1 and 100 s^−1^.

### 2.9. Fourier Transform Infrared (FTIR) Spectra

FTIR spectra were recorded to understand the effect of NBs on the secondary structures of the protein. FTIR spectra (Thermo Electron Corporation, Madison, WI, USA) were recorded from 4000 to 600 cm^−1^. Each spectrum was recorded as an average of 64 scans at a resolution of 4 cm^−1^. For each measurement, 5 µL of the sample was placed in the sample chamber, and the sample spectra were collected. Before starting the sample measurements, a time background was collected. The data was collected using OMNIC v32 software (Thermo Fisher Scientific Inc., Waltham, MA, USA) in the system [17].

### 2.10. Sodium Dodecyl Sulphate–Polyacrylamide Gel Electrophoresis (SDS-PAGE)

SDS-PAGE was run using the Bio Rad Laboratories (Hercules, CA, USA) MiniProtean Tetra cell system with 4–20% TGX precast gels. Each sample was diluted 1:1 with sample Lamelli buffer from Bio-Rad Laboratories, followed by heating to 95 °C for 5 min. Retentate samples (at 120 min) were diluted with water 1:1 before adding the sample buffer. For reducing SDS-PAGE, 2.5% of β-mercaptoethanol was added to the Lamelli buffer. The electrophoresis running tank was filled with running buffer (pH 8.3), and then 10 μL samples were loaded into the gels. The gels were run at 100 V until the end of the run. Then the gels were fixed in methanol and acetic acid (40:10) solution made up to 100 mL using water for 15 min. The gels were washed in water twice, followed by staining in Coomassie blue (G 250) from Bio Rad overnight. The gels were destained in water for three hours, changing the water three times.

### 2.11. Statistical Analysis

Statistical analysis was conducted using OriginPro software (Origin 2024b 10.15, OriginLab Corporation, Northampton, MA, USA) with a two-way ANOVA at α = 0.05% significance level. Each trial was repeated three times for both the control and nanobubble-treated whey. Each analysis was performed at least in duplicate measurement replicates, and all the data were represented as mean ± standard error.

## 3. Results

### 3.1. Changes in Permeation Flux of Model Cheese Whey

Membrane permeation flux, a crucial parameter determining the efficiency of membrane processes, usually declines over time due to concentration polarization or membrane fouling on the surface, as indicated in Figure 1. Generally, filtration time and permeation flux are inversely proportional. The air nanobubble-treated model cheese whey (NBW) displayed a significant increase (*p* < 0.05) in UF membrane flux compared to the control whey (CW), between 20 and 60 min within the 120 min total filtration runtime (Figure 2). The presence of nanobubbles in NBW, in the initial stages of filtration, increases the flux by breaking up aggregates at the membrane surface and thereby delaying the increase in the thickness of the concentrated polarization layer at the membrane surface [11,14]. Furthermore, there was a reduction in flux over time, until there was no difference between NBW and CW at 80 min, indicating the decreased influence of NBs, owing to their implosion at the membrane surface and depletion over time. In our study, nanobubbles were generated before UF, and a decay in NB concentration is expected with increasing run time for UF, which is also reported by Li et al. [13] for micro-nanobubbles (MNB), wherein a decrease was observed in the count of MNBs with an increase in UF run time for water and humic acid with MNBs. Overall, the incorporation of NBs in the feed before UF increases the peak permeation flux by 34 ± 5%, which is an indicator of membrane filtration efficiency.

### 3.2. Changes in Composition and Concentration Factor of Model Cheese Whey

The total solids and protein contents of UF retentate were monitored at 20 min intervals during the 120 min UF run time. After UF filtration for 120 min, the total solid content significantly increased (*p* < 0.05) for the final retentate of CW and NBW to 8.48% and 10.86% compared to the 5.12% at 0 min time, respectively (Figure 3). A similar trend was observed for protein content. The initial protein content for both CW and NBW was 0.94%, which significantly increased to 3.1% and 4.07% (*p* < 0.05), respectively. NBW demonstrated significantly higher total solids and protein content at 100 and 120 min (*p* < 0.05), among the highest total solids and protein observed at all time points, compared to control CW. The increase in total solids and protein content in NBW correlates well with increased flux and a higher concentration factor, leading to faster concentration and retention of solids. Specifically, a linear significant positive correlation (*p* < 0.001) was observed for concentration factor with total solids (r = 0.98) and protein (r = 0.97).

The observed increase in total solids and protein content after the ultrafiltration of cheese whey is as expected from the UF process [1,4]. Additionally, ash content measured at 0 min and 120 min showed no significant difference (*p* > 0.05) between control CW and NBW at 0 and 120 min (Table 1).

The UF permeate from NBW treatment exhibited similar compositional characteristics compared to CW permeate (Table 2), at the 120 min time point (*p* > 0.05). As expected, the permeate composition is similar for the NB-treated and control samples at 120 min of UF. This similarity in permeate composition towards the end of filtration also confirms that the effect of NBs in the UF of cheese whey was more prominent at the earlier stages of filtration until the depletion of the NBs, which is also evident from the increased flux until 60 min compared to CW.

Concentration factor (CF) is the ratio of the initial amount (weight was measured for our study) of whey used as feed to the amount of retentate after filtration or at specific time points, as indicated in Figure 2. The CF at 120 min was 4.5× for NBW compared to 3.2× for CW (*p* < 0.05). A higher CF was consistently observed for NBW at all time points compared to CW. The increase in flux at the earlier stages of filtration (until 80 min) contributes to a higher concentration factor by allowing faster removal of permeate (Figure 2). The higher total solids and protein content in NBW, owing to higher permeation flux, is correlated well with a significant increase in concentration factor during UF, allowing faster concentration, retaining higher solids and protein.

At 120 min, a lower flux was observed for NBW compared to CW, owing to the significant increase in total solids and protein levels to 10.9% (*w*/*w*) and 4.08% (*w*/*w*) in NBW compared to CW (8.49% and 3.1% (*w*/*w*)), respectively (Figure 2 and Figure 3). An increase in total solids and protein content in retentate increases the viscosity of NBW and the concentration polarization layer on the UF membrane surface, thereby lowering flux at 120 min. This phenomenon is consistent with other studies reporting a higher accumulation of solute materials on the membrane surface with an increase in the volume concentration factor and total solids of the feed, leading to an increase in the thickness of the polarized layer and subsequent reduction in flux [1,8]. Usually, the major foulants on the membrane surface have been milk proteins; for instance, Tong et al. [18] found that 95% of foulants on the membrane surface were milk proteins during ultrafiltration of whole milk. Regardless, the presence of fat in the feed (0.2% *w*/*w*) could also cause some pore blockage, leading to resistance in permeation at higher CF (4.5×) for NBW, as reported for fresh cheese whey during ultrafiltration [8].

Overall, these results indicate that nanobubble incorporation increased the concentration of protein and total solids in the retentate by 33 ± 10% and 28 ± 5% at 120 min, associated with 34 ± 5% higher peak flux and 40 ± 4% higher concentration factor (at 120 min) in NBW compared to the control (CW). Furthermore, a higher permeation of lactose and minerals was observed in permeate from NBW after UF, which also correlates with higher flux, concentration factor, and filtration efficiency.

### 3.3. Viscosity

Nanobubble generation in the model cheese whey did not affect the viscosity of the NBW feed stream at 0 min (*p* > 0.05). The UF retentate from NBW showed a significantly (*p* < 0.05) higher increase in viscosity from 0 min to 120 min of UF compared to CW (Table 1). This observation aligns with the higher total solids and protein content in NBW compared to CW, at 120 min. Furthermore, the higher viscosity of NBW at 120 min suggests higher resistance to flow, contributing to the decrease in flux observed at 120 min (Figure 2).

Previously, nanobubbles had been associated with a reduction in viscosity for high protein dispersions [14,19]. This effect was not observed in our study, possibly owing to bubble implosion at the concentration polarization layer during the initial phases of UF, which led to depletion in concentration of nanobubbles towards the end of filtration. This phenomenon is also evident from the peak flux observed at 20 min of UF and the later decline in flux until 120 min.

### 3.4. Fourier Transform Infrared (FTIR) Spectra and SDS-PAGE

An important aspect during processing or any processing treatment is preserving the functionality of whey proteins, as any processing can lead to structural changes, thereby affecting functional characteristics of whey proteins [20]. FTIR spectra were recorded to assess the impact of NB treatment on the secondary structures of whey proteins (Figure 4), and no differences were observed in the FTIR spectra between the control and NB-treated whey in the amide I (1600–1700 cm^−1^) and amide II region (1500–1600 cm^−1^) (Figure 4). This indicated that nanobubble treatment did not alter the secondary structures of whey proteins present in model cheese whey.

SDS-PAGE was performed in non-reducing (Figure 5A) and reducing conditions (Figure 5B). There was no difference observed among CW and NBW samples at 0 and 120 min, indicating that NBs did not cause any significant change to whey protein structures before UF and at the end of UF, which corroborated our FTIR findings. There were evident increases in the protein concentration and aggregate formation with UF, for both CW and NBW, with both treatments exhibiting the same changes. 

The bands of individual whey proteins were darker and thicker in retentate samples (120 min) compared to initial samples at 0 min, indicating their increasing concentration during UF. This supports the composition and viscosity changes in the study. There was no whey protein present in permeate for both CW and NBW, supporting the low total protein values of permeate from both streams (0.06% protein) and the retention of almost all of the whey proteins in the retentate during the UF process. Overall, NB treatment did not have any major effect on individual whey proteins as indicated by SDS-PAGE and FTIR findings. Therefore, the NB incorporation enhanced the efficiency of UF by reducing concentration polarization/fouling during filtration and did not affect the protein aggregates in the concentrated permeate.

### 3.5. Particle Size and Zeta Potential

The incorporation of NB into the feed did not induce any significant change in the particle size of whey protein aggregates compared to the control (CW) after UF (*p* > 0.05) (Figure 6). However, particle size increased significantly (*p* > 0.05) after ultrafiltration (UF) at 120 min for both CW and NBW. Specifically, the peak particle size (mode values) increased from 332 and 295 in CW, to 503 and 550 in NBW at 0 min and 120 min, respectively. Previously, the particle size mode values have been reported between 250 and 375 nm for cheese whey, consistent with our results [3,8,23]. The increase in particle size at 120 min in NBW and CW corresponds to the increase in total solids and protein content, wherein aggregation among whey proteins with concentration would increase particle size [8]. Overall, NBs did not significantly impact the particle size of whey proteins in NBW compared to CW, suggesting that the NB treatment did not induce notable changes in the size distribution of whey protein aggregates before and during the ultrafiltration process.

The nanobubble treatment in NBW (−14.75 mV) did not induce any significant change in zeta potential compared to the control cheese whey (CW) (−14.54 mV) feed for UF (Figure 7). However, a decrease was observed in the absolute zeta potential for the retentate at 120 min in both samples compared to the zeta potential at 0 min (*p* < 0.05), indicating the lower stability of the larger aggregates with UF. This decrease in zeta potential after UF at 120 min was more pronounced (*p* < 0.05) in NBW (−9.4 mV), as compared to CW (−11 mV). These findings suggest that the increased efficiency of filtration induced concentration caused a decrease in absolute zeta potential, which could be because of two possible reasons. Firstly, the aggregation of whey protein, as observed by an increase in particle size, could lead to a decrease in absolute zeta potential values [24,25]. Secondly, the decrease in zeta potential could be attributed to a change in the dynamic of the minerals and higher concentrations of minerals on the particle stability during NBW ultrafiltration (UF), causing changes in surface charge [25]. This was also supported by conductivity and higher ash content findings, wherein NBW had slightly higher conductivity (2.79 mS/cm) and ash content (0.33%) compared to CW (2.72 mS/cm and 0.30%) at 120 min (Table 1), albeit not significant. Higher ionic strength can lead to compression at the electric double-layer surface of the proteins and affect the zeta potential [25].

Additionally, a similar trend was also observed for NBW permeate ash content and zeta potential, although the changes were not significant (*p* > 0.05).

### 3.6. Proposed Mechanism of Action of Nanobubbles During UF of Model Cheese Whey

The specific mechanism of nanobubbles in preventing deposition of milk solids or a layer on the membrane surface can be twofold (Figure 8). Initially, the nanobubbles are relatively stable in the initial model cheese whey system, together with the dispersed small whey protein aggregates and fat globules, owing to their negative charges (Figure 8). Concentration polarization occurs with the increase in concentration at the surface of the filter as the filtration progresses, leading to increased interaction and aggregation of proteins and fat. This concentrated layer is affected to a lesser and lesser extent over time by the turbulence and shear forces created by the fluid flow in the membrane, leading to fouling. NBs inhibit the interactions of proteins and fat globules among themselves, in turn reducing their electrostatic interaction with these particles, which reduces the rate of their aggregation. The second mechanism of action is the turbulence, shear forces, and cavitation produced at the membrane surface (Figure 8), owing to the implosion of the NBs, which disrupts the aggregates and the concentration polarization layers formed.

These phenomena are supported by our findings in this work, wherein the 34 ± 5% highest peak flux was observed at 20 min for NBW compared to CW. This indicates the prevention of concentration polarization at the UF membrane surface by the action of the NBs in the initial stages of the UF. This increase in flux for NBW led to a 40 ± 4% higher concentration for NBW as compared to CW factor at the end of UF (120 min), increasing the efficiency of filtration. This increase in CF and flux resulted in the amount of protein and total solids in the retentate being higher by 33 ± 10% and 28 ± 5% at 120 min for NBW compared to CW. No significant difference in particle size, zeta potential and SDS-PAGE, and FTIR was observed for the control and nanobubble-treated model cheese whey feed, indicating that nanobubbles did not affect the particle sizes in the initial feed containing whey protein aggregates (and a very small concentration of fat globules).

The second aspect of the mechanism of action of NBs occurs owing to the implosion of the bubbles at the filtration surface, supported by the depletion of bubbles over time, evident from the decrease in flux for NBW at 120 min, as compared to the highest peak flux at 20 min. Other studies have reported the effect of electrostatic hindrances and bubble implosion as the main reasons for the breaking up of protein aggregates in high-concentration dispersions, and the combined effects have been reported in the UF of nanobubble-treated skim milk dispersions and milk protein concentrates [12,14,19]. Only one study has used NBs for the UF of a 25% total solid feed produced by hydrating non-fat dry milk, which was maintained at the same concentration for 1 h (by recirculation after adding back the permeate to the feed). They reported significantly higher permeation flux for the NBs incorporated UF process at every time point and a lower thickness of the fouling layer on the membrane at the end of UF compared to the control UF process [14]. This study further indicates the promise of nanobubble technology in membrane filtration; however, the constant concentration mode UF process used is not commercially scalable.

Overall, our findings indicate a significant increase in flux due to the presence of nanobubbles in NBW by creating electrostatic hindrances and implosion-induced turbulence and shear forces at the surface of the membrane in the initial phases of the ultrafiltration, prior to NB depletion in the system over time during UF. Hence, the incorporation of NBs into UF will be a great advantage to the dairy industry, helping reach target solids in less production time. In our study, a concentration factor of 3× was reached for retentate with NB treatment at 79.77 ± 1.23 min, compared to at 111.40 ± 1.84 min for the control, which indicates a processing time reduction of 28.27% ± 2.33 to reach the same concentration factor as the control. Our work primarily focused on evaluating the technical feasibility and performance of NBs during ultrafiltration at the laboratory scale. Future studies should focus on understanding the interactions, specifically at the molecular level, between whey proteins and nanobubbles, including the interactions at the membrane surface during UF. Additionally, there is a need for pilot and industrial-scale studies to be conducted to enable the commercial scaling-up of nanobubble application in membrane filtration, including techno-economic and energy assessments, as well as compatibility assessments for different membrane materials, pore sizes, and streams.

## 4. Conclusions

In this study, the application of nanobubbles (NBs) in whey ultrafiltration yielded a notable increase in total solids (28%) and protein (33%) in the final retentate, within the same amount of processing time as control ultrafiltration without NBs. Permeation flux was substantially enhanced by NBs, particularly up to 60 min, resulting in a higher concentration factor (40%) towards the end of UF compared to the control. This led to a 28.27% reduction in processing time to reach the same concentration factor as the control. The results suggest that the increase in flux and concentration of total solids and protein during UF of whey is due to the electrostatic hindrance and bubble implosion associated with shear and turbulence at the membrane surface by NBs. These phenomena delay the formation of a polarized layer on the membrane surface, inhibit fouling, and improve filtration efficiency. This is the first time that in a mixed system, it is being observed that nanobubbles can have prolonged effects until complete depletion, even if they are not being actively and continuously generated in the system. Furthermore, the increase in protein and total solids content in the NB-generated cheese whey (NBW) for the same processing time as the control CW implies potential time and energy savings, which can also be applied to diafiltration in conjunction with ultrafiltration. Furthermore, the continuous generation and incorporation of NBs during the filtration process can reduce concentration polarization and improve filtration efficiency. Overall, nanobubble technology is a very promising technology for the improvement of membrane filtration of different streams for the food and dairy industry, with potential to achieve higher concentrations and shorter run times.

## Figures and Tables

**Figure 1 foods-14-04224-f001:**
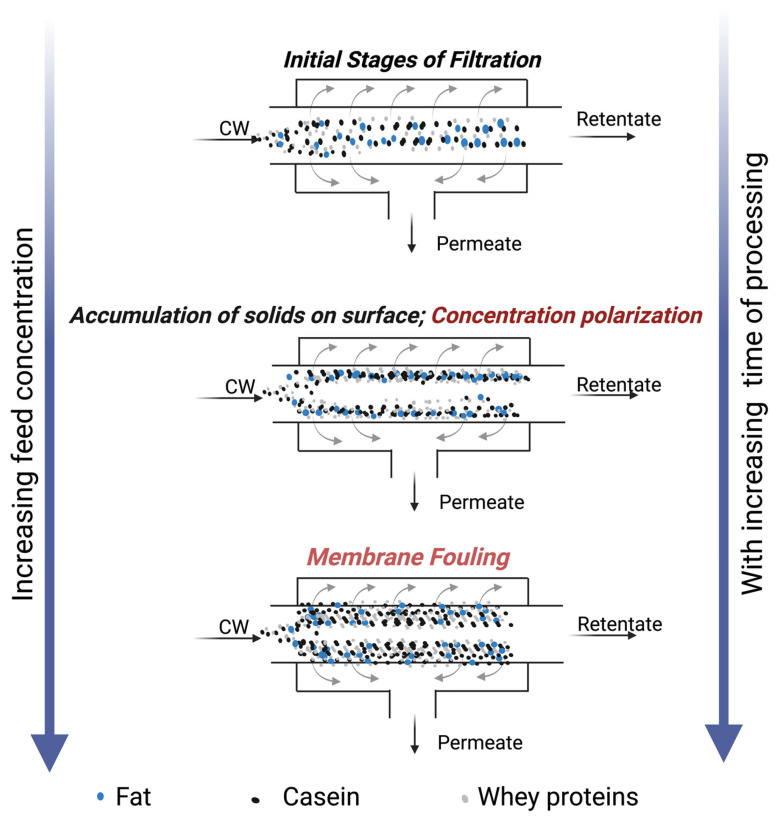
Illustration of concentration polarization and membrane fouling at various stages of membrane processing of model cheese whey.

**Figure 2 foods-14-04224-f002:**
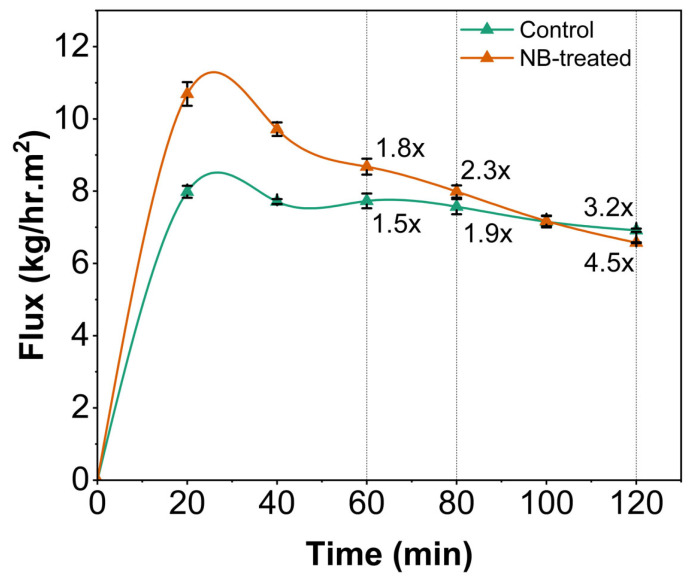
Permeation flux (kg/h·m^2^) over time during ultrafiltration of model cheese whey (control and NB-treated) for 120 min. x represents the concentration factor and is only represented at significant increases using dotted lines at specific time points.

**Figure 3 foods-14-04224-f003:**
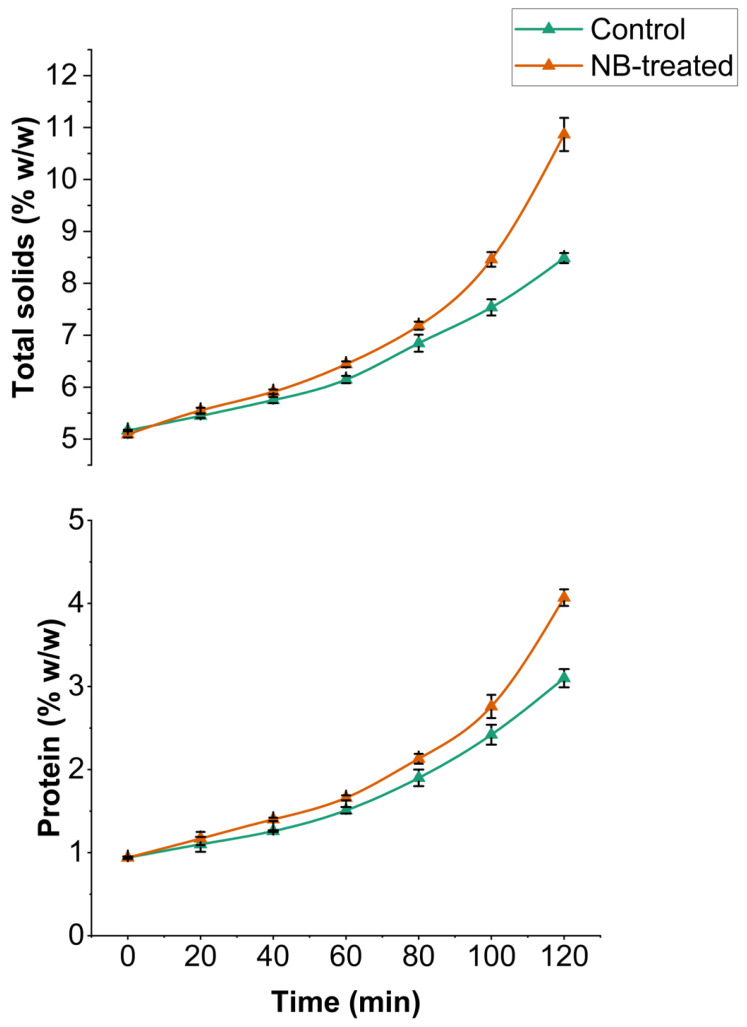
Changes in total solids and protein content (% *w*/*w*) during ultrafiltration of model cheese whey retentate (control and NB-treated) for 120 min.

**Figure 4 foods-14-04224-f004:**
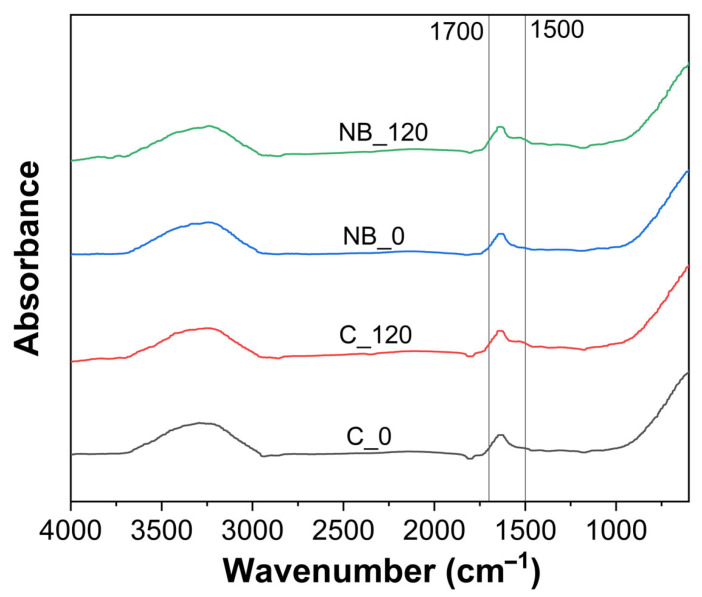
FTIR spectra before (0 min) and after (120 min) ultrafiltration of control (C_0 and C_120) and NB-treated (NB_0 and NB_120) model cheese whey.

**Figure 5 foods-14-04224-f005:**
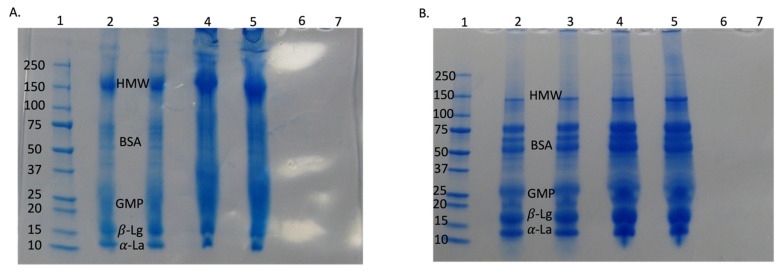
SDS-PAGE for control and NB-treated whey under non-reducing conditions (**A**) and reducing conditions (**B**). Lane 1—standard protein ladder; Lanes 2 and 3 are CW and NBW, respectively, at 0 min (initial); Lanes 4 and 5 are CW and NBW, respectively, at 120 min (final retentate); Lanes 6 and 7 are CW permeate and NBW permeate. HMW—high molecular weight aggregates; BSA—bovine serum albumin; GMP—glycomacropeptide [21,22]; α-La—alpha lactalbumin; β-Lg—beta-lactoglobulin.

**Figure 6 foods-14-04224-f006:**
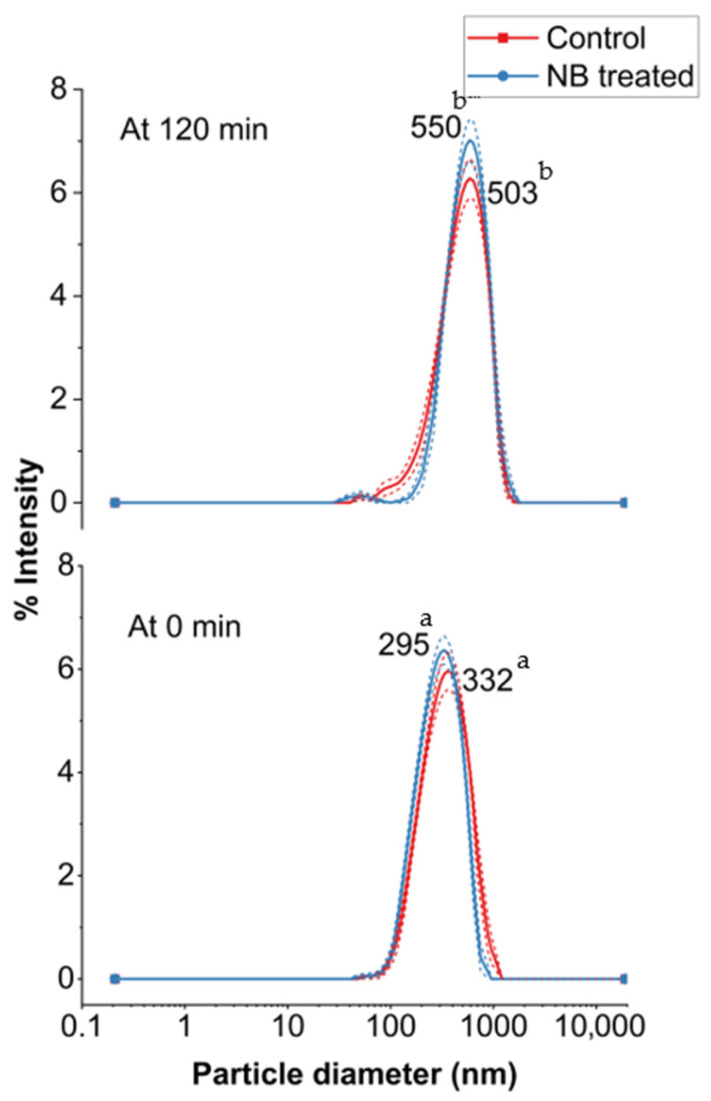
Particle size distribution (nm) before (0 min) and after (120 min) ultrafiltration of control (C_0 and C_120) and nanobubble-treated (NB_0 and NB_120) model cheese whey. ^a,b^ Values not sharing a common lowercase superscript are significantly different (*p* < 0.05).

**Figure 7 foods-14-04224-f007:**
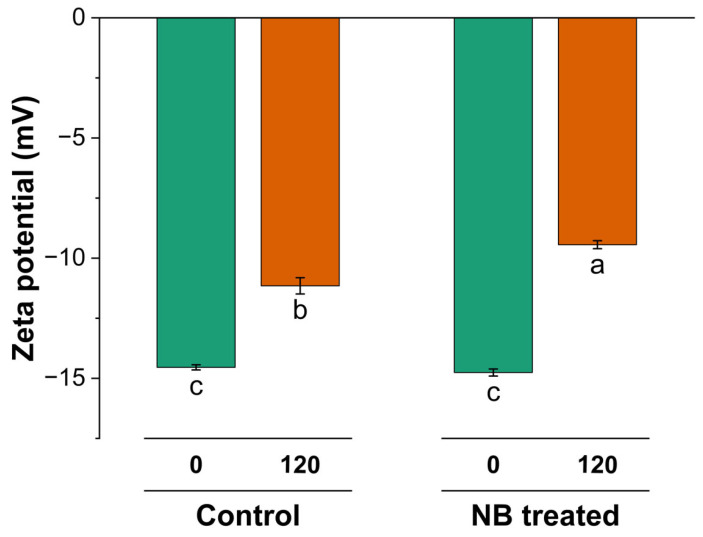
Zeta potential (0 min) and after (120 min) ultrafiltration of control (C_0 and C_120) and NB-treated (NB_0 and NB_120) model cheese whey. ^a–c^ Values not sharing a common lowercase superscript are significantly different (*p* < 0.05).

**Figure 8 foods-14-04224-f008:**
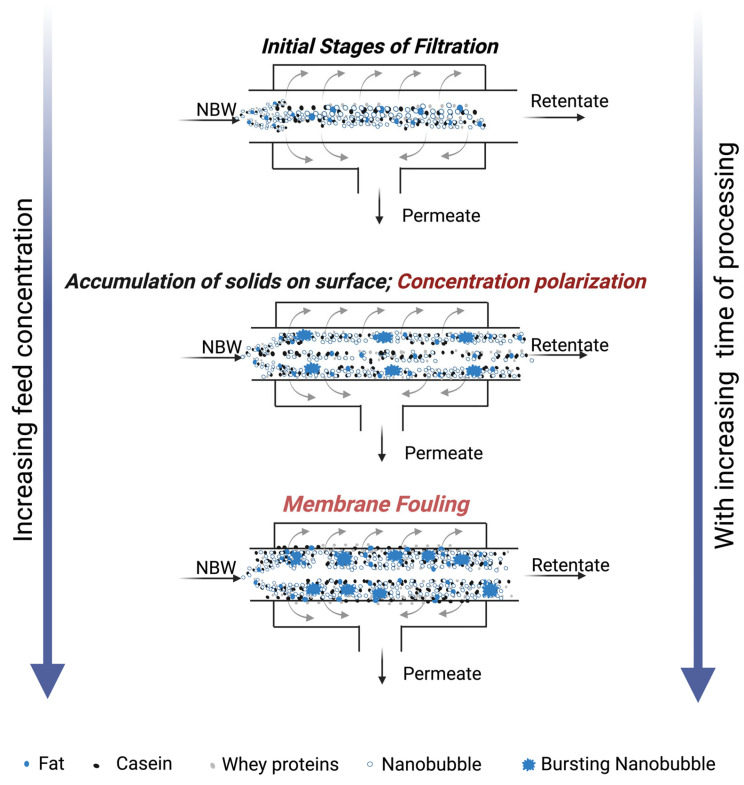
Illustration of the mechanism of nanobubbles alleviating concentration polarization and membrane fouling at various stages of membrane processing of nanobubble-treated model cheese whey (NBW).

**Table 1 foods-14-04224-t001:** Physico-chemical analysis of model cheese whey (control (CW) and nanobubble-treated (NBW)) at 0 min (initial feed) and at 120 min (final retentate) after ultrafiltration run.

Parameter	Time	CW	NBW
Total solid (% *w*/*w*)	At 0 min (Initial)	5.16 ± 0.01 ^a^	5.09 ± 0.06 ^a^
	At 120 min (Final)	8.48 ± 0.09 ^a^	10.86 ± 0.32 ^b^
Protein (% *w*/*w*)	At 0 min (Initial)	0.94 ± 0.01 ^a^	0.94 ± 0.01 ^a^
	At 120 min (Final)	3.10 ± 0.11 ^a^	4.07 ± 0.10 ^b^
Ash (% *w*/*w*)	At 0 min (Initial)	0.21 ± 0.01 ^a^	0.19 ± 0.02 ^a^
	At 120 min (Final)	0.30 ± 0.05 ^b^	0.33 ± 0.01 ^b^
Viscosity (m·Pa·s) At 100 s^−1^	At 0 min (Initial)	2.05 ± 0.01 ^a^	2.00 ± 0.00 ^a^
	At 120 min (Final)	2.57 ± 0.09 ^b^	3.24 ± 0.10 ^a^
Conductivity (mS/cm)	At 0 min (Initial)	2.37 ± 0.01 ^a^	2.38 ± 0.01 ^a^
	At 120 min (Final)	2.72 ± 0.03 ^a^	2.79 ± 0.03 ^a^

^a,b^ Values in a row not sharing a common lowercase superscript are significantly different (*p* < 0.05). Values in a column for a specific parameter are significantly different (*p* < 0.05) from each other, indicating a significant increase was observed in all parameters at 120 min from 0 min samples.

**Table 2 foods-14-04224-t002:** Physico-chemical analysis of permeate from control (CW) and nanobubble-treated whey (NBW) after ultrafiltration for 120 min.

Parameter	CW Permeate	NBW Permeate
Total solids (% *w*/*w*)	3.66 ± 0.04 ^a^	3.83 ± 0.05 ^a^
Protein (% *w*/*w*)	0.06 ± 0.01 ^a^	0.06 ± 0.06 ^a^
Ash (% *w*/*w*)	0.13 ± 0.08 ^a^	0.16 ± 0.05 ^b^
Lactose (% *w*/*w*) ^1^	3.46 ± 0.04 ^a^	3.61 ± 0.05 ^a^
Zeta potential (mV)	−16.15 ± 0.56 ^a^	−16.38 ± 0.57 ^a^

^a,b^ Values in a row not sharing a common lowercase superscript are significantly different (*p* < 0.05). ^1^ Lactose was estimated by the difference method (lactose = total solids − (ash + protein)), assuming that fat present in the initial whey (0.2% *w*/*w*) would be retained in the retentate.

## Data Availability

The original contributions presented in the study are included in the article, further inquiries can be directed to the corresponding author.

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
