# Peer review of "Effect of Nanobubbles on Ultrafiltration Membrane Performance and Properties of Model Cheese Whey"

_foods, 2025, doi:10.3390/foods14244224_

Round 1

Reviewer 1 Report

Comments and Suggestions for Authors

This paper submitted for review assessed the impact of air-generated nanobubbles on ultrafiltration membrane performance and the physicochemical properties of model cheese whey. This study is relevant as it aims to determine whether nanobubbles can mitigate membrane fouling and enhance UF efficiency in dairy processing applications. However, I believe there are several points that should be addressed to improve the quality of the manuscript:

  1. It would be beneficial to begin the abstract with a brief background that clarifies the challenge or motivation behind introducing nanobubbles to give the reader some context before describing what the study investigated.
  2. The results presented in the abstract were majorly descriptive without context. It would be helpful to include short quantitative indicators such as percentage improvement or at least highlight the practical significance of the observation. Also, the claims in the concluding sentence are too strong and slightly speculative. This could be moderated to “nanobubble technology shows promising potential to improve membrane filtration…”.
  3. The introduction section contains too much textbook-level information such as the paragraphs describing whey composition, membrane types, UF cutoffs, definitions of fouling, etc. Although these are correct, they do not directly support the paper’s specific research problem. I suggest that the authors condense this general information and focus instead on important discussions such as why UF performance is a challenge, how fouling specifically affects whey processing, what is missing in current mitigation strategies and how this study addresses that.
  4. Several ideas appear multiple times in the same section which affects readability. For example, UF is widely used in dairy is repeated twice (lines 33–38 and 43–48), membrane fouling reduces flux is repeated in lines 50–53 and 59–63, nanobubbles improve flux and reduce fouling is mentioned at least 3 times. It would be helpful to eliminate these duplicated statements and keep one strong, well-referenced explanation for each concept.
  5. While the introduction states that nanobubbles improve flux and reduce fouling, it does give any explanation on how they do it. It would be helpful to include 2–3 mechanistic sentences to justify why NBs could improve UF performance. Also add a clear hypothesis statement.
  6. The authors conclude the introduction by saying that “there is no study available that understands the effect of nanobubbles on ultrafiltration of cheese whey”. However, earlier statements imply NB effects in milk and other dairy systems, so why might whey specifically behave differently? does whey’s composition (lactose, salts, proteins) significantly connect to NB behavior? I suggest that the authors highlight these (why whey protein systems are unique and why NB performance needs to be tested in whey, not assumed from milk or model proteins) to strengthen the scientific justification of the study.
  7. Line 159: The double parentheses should be removed.
  8. Line 221: The significance level was incorrectly stated. It should be “α = 0.05,” not “0.05%.”
  9. Generally, many paragraphs in the results section only restate numerical results that have already been presented in Figures/Tables without deeper mechanistic or statistical interpretation. For example, in section 3.2, the authors only repeated the data from Figures/Tables and concluded that “the observed increase in total solids and protein content after ultrafiltration of cheese whey is consistent with findings from other studies”, without stating the significance of the observed trends in the context of the present study. Beyond saying that higher flux enables faster concentration, why does NBW enhance solids retention?
  10. Statements about concentration polarization and fouling are repeated across multiple sections and should be condensed to reduce redundancy and focus specifically on how NB treatment alters these processes.
  11. The manuscript claims that NB treatment “increases removal of lactose and ash,” but the differences (3.61% vs. 3.47%) are small and statistically insignificant. The authors should explicitly acknowledge when differences are not significant and provide justifications if any.
  12. Line 344: Much of the SDS-PAGE discussion repeats known properties of WPC processing rather than analyzing how NB treatment affects protein stability. I suggest that the authors focus the discussion on comparisons between CW and NBW, not on generic whey protein behavior.
  13. Line 467: The statement that NB treatment could reduce processing time by “approximately 30%” is an overstatement that should be moderated unless validated at industrial scale.

Author Response

Reviewer 1

Response

This paper submitted for review assessed the impact of air-generated nanobubbles on ultrafiltration membrane performance and the physicochemical properties of model cheese whey. This study is relevant as it aims to determine whether nanobubbles can mitigate membrane fouling and enhance UF efficiency in dairy processing applications. However, I believe there are several points that should be addressed to improve the quality of the manuscript:

Thanks for your feedback and valuable suggestions. We have made changes based on suggestions.

It would be beneficial to begin the abstract with a brief background that clarifies the challenge or motivation behind introducing nanobubbles to give the reader some context before describing what the study investigated.

We have included brief background in the beginning of the abstract. (L 9-11)

The results presented in the abstract were majorly descriptive without context. It would be helpful to include short quantitative indicators such as percentage improvement or at least highlight the practical significance of the observation. Also, the claims in the concluding sentence are too strong and slightly speculative. This could be moderated to “nanobubble technology shows promising potential to improve membrane filtration…”.

We have made suggested changed and included quantitative indicators in the abstract. (L17, 23, 26)

Also, revised the concluding sentence as suggested. (L 27)

The introduction section contains too much textbook-level information such as the paragraphs describing whey composition, membrane types, UF cutoffs, definitions of fouling, etc. Although these are correct, they do not directly support the paper’s specific research problem. I suggest that the authors condense this general information and focus instead on important discussions such as why UF performance is a challenge, how fouling specifically affects whey processing, what is missing in current mitigation strategies and how this study addresses that.

We have condensed the information to make the introduction focused on the background, problem statement, hypothesis and objectives.

Several ideas appear multiple times in the same section which affects readability. For example, UF is widely used in dairy is repeated twice (lines 33–38 and 43–48), membrane fouling reduces flux is repeated in lines 50–53 and 59–63, nanobubbles improve flux and reduce fouling is mentioned at least 3 times. It would be helpful to eliminate these duplicated statements and keep one strong, well-referenced explanation for each concept.

Removed the duplicate statements from the introduction and made it concise.

While the introduction states that nanobubbles improve flux and reduce fouling, it does give any explanation on how they do it. It would be helpful to include 2–3 mechanistic sentences to justify why NBs could improve UF performance. Also add a clear hypothesis statement.

We have included the mechanism for explaining the effect of nanobubbles (L 67-74) and also added the hypothesis. (L 76-79)

The authors conclude the introduction by saying that “there is no study available that understands the effect of nanobubbles on ultrafiltration of cheese whey”. However, earlier statements imply NB effects in milk and other dairy systems, so why might whey specifically behave differently? does whey’s composition (lactose, salts, proteins) significantly connect to NB behavior? I suggest that the authors highlight these (why whey protein systems are unique and why NB performance needs to be tested in whey, not assumed from milk or model proteins) to strengthen the scientific justification of the study.

Thanks for your comment.

There is only one study conducted with NBs for concentration of skim milk using UF, which focused on constant concentration mode (adding permeate back to concentrate at regular intervals) with UF of skim milk at fixed total 25% solids. This study showed the promise of nanobubble technology, however the constant concentration mode UF process used is not scalable (L 453-460).  However, our study is the first to perform UF for cheese whey starting with initial composition and reaching to final solids in retentate to understand the fate of nanobubbles during concentration using UF. We have included a research gap statement in the introduction (L 71-75)

The reason for choosing whey as a stream was that the cheese whey is most significant by product of cheese industry and all dairy processing industries utilizes UF to concentrate cheese whey. Therefore, it is important to understand the effect of nanobubbles on membrane efficiency and properties of cheese whey during UF. (L 70-74)

Line 159: The double parentheses should be removed.

Removed. Thanks for pointing it out.

Line 221: The significance level was incorrectly stated. It should be “α = 0.05,” not “0.05%.”

We have changed it as suggested. (L 226)

Generally, many paragraphs in the results section only restate numerical results that have already been presented in Figures/Tables without deeper mechanistic or statistical interpretation. For example, in section 3.2, the authors only repeated the data from Figures/Tables and concluded that “the observed increase in total solids and protein content after ultrafiltration of cheese whey is consistent with findings from other studies”, without stating the significance of the observed trends in the context of the present study. Beyond saying that higher flux enables faster concentration, why does NBW enhance solids retention?

We have improved the discussion in the manuscript. We have added in section 3.2 on why the NBW had higher total solids and protein.

The statement “the observed increase in total solids and protein content after ultrafiltration of cheese whey is consistent with findings from other studies”, indicates the expected effect of UF and is not associated with the NB treatment effect.

Furthermore, we have explained in detail the mechanistic action of NBs in section 3.6, based on our observations.

Statements about concentration polarization and fouling are repeated across multiple sections and should be condensed to reduce redundancy and focus specifically on how NB treatment alters these processes.

Thanks for your comment.

However, these statements are used in different sections to clearly state the perspectives of the discussion. We have correlated the results from the study with fouling in each section since that was the major objective of the study.

The manuscript claims that NB treatment “increases removal of lactose and ash,” but the differences (3.61% vs. 3.47%) are small and statistically insignificant. The authors should explicitly acknowledge when differences are not significant and provide justifications if any.

We have removed this statement and rephrased it in the manuscript, not focusing on insignificant results.  (L280-285)

Line 344: Much of the SDS-PAGE discussion repeats known properties of WPC processing rather than analyzing how NB treatment affects protein stability. I suggest that the authors focus the discussion on comparisons between CW and NBW, not on generic whey protein behavior.

Thank you for your inputs. We appreciate and agree to this.

We have already stated that no differences were observed between CW and NBW in the beginning of the section. (L 377-379)

Also, we have removed the discussion on the protein changes associated with WPC processing, in general, and have focused on stating the difference between CW and NBW.

Line 467: The statement that NB treatment could reduce processing time by “approximately 30%” is an overstatement that should be moderated unless validated at industrial scale.

Thanks for your comment. We have included that based on calculation from results in our study. We have included the exact numbers with standard error by calculating it from concentration factor data. Further, included the time taken for CW and NBW to reach 3x, to make it clear (L466-469).  To emphasize the relevance of scale up, we have included the statement below:

Additionally, there is need for conduct pilot and industrial scale studies to enable commercial scaling up of nanobubbles application in membrane filtration, including techno-economic and energy assessments, compatibility assessment for different membrane materials, pore sizes and streams.

Also, included other potential futuristic work that can be conducted based on other reviewer suggestions.

Reviewer 2 Report

Comments and Suggestions for Authors

This study investigates the application of nanobubbles (NBs) in the ultrafiltration (UF) process of cheese whey, systematically examining for the first time the effects of NBs on membrane permeate flux, concentration factor, physicochemical properties of whey, and protein structure. The core findings—such as a 34% increase in peak flux, a 40% improvement in concentration factor, and no alteration of whey protein secondary structure—provide a novel technological pathway for optimizing membrane filtration processes in the dairy industry. The research perspective is innovative, filling a gap in the field and overcoming the limitations of conventional membrane fouling control techniques (e.g., ultrasound, gas sparging). This work is the first to apply NB technology to the cheese whey UF process, demonstrating that NBs can significantly enhance filtration efficiency by mitigating concentration polarization and suppressing membrane fouling. The experimental design is systematic and well-supported by data, employing a multidimensional research framework that integrates “process performance, physicochemical properties, and structural characterization.” UF experiments monitored changes in permeate flux and concentration factor, complemented by physicochemical analyses (total solids, protein content), and protein structural stability was validated using FTIR, SDS-PAGE, particle size, and zeta potential measurements. The study is application-oriented with significant industrial value, addressing critical issues such as flux decline and increased energy consumption caused by membrane fouling in dairy UF processes. NB technology can shorten the time required to achieve the target solid content in whey concentration by 30%, offering both energy-saving and high-efficiency advantages. However, the manuscript has several issues that need systematic improvement to enhance its scientific rigor and translational value.

  1. The mechanism description is overly general: It only proposes two pathways—“electrostatic hindrance” and “turbulence and shear force generated by bubble rupture”—without clarifying the specific interactions between NBs and whey foulants such as proteins and fats (e.g., hydrogen bonding, hydrophobic interactions), making it difficult to explain why NBs can selectively inhibit whey protein deposition on the membrane surface.
  2. Lack of microscopic characterization evidence: There is no microscopic observation of the membrane fouling layer (e.g., SEM, AFM), so the direct impact of NBs on fouling layer thickness and density cannot be confirmed, nor is there a quantitative analysis of the shear force generated by NB rupture.
  3. Missing correlation between NB properties and effects: Key parameters of the generated NBs (e.g., particle size distribution, concentration, zeta potential, stability) are not characterized, making it impossible to establish a quantitative relationship between NB properties and their effects, such as the impact of different NB sizes or concentrations on flux improvement.
  4. Ambiguity in NB generation process parameters: The manuscript only mentions “oxidizer circulation for 10 minutes, air pressure at 20 PSI, flow rate of 0.25 LPM,” without specifying critical conditions such as liquid temperature and pH during NB generation, nor does it verify the uniform dispersion of NBs in whey after generation.
  5. Unclear membrane pretreatment and cleaning procedures: The methods for membrane pretreatment prior to UF experiments (e.g., soaking, cleaning steps), calibration data for initial membrane flux, and post-experiment cleaning and regeneration procedures are not described, affecting experimental repeatability.
  6. Failure to exclude potential confounding factors: The study does not analyze the effects of NBs on the migration behavior of lactose and minerals in whey, nor does it verify whether NBs alter the membrane’s retention characteristics (e.g., differences in retention rates for proteins of varying molecular weights).
  7. Lack of consideration for the complexity of real whey: The experiments use model whey (WPC + lactose + water) with a simple composition, without simulating the complex matrix of actual cheese whey (e.g., free fatty acids, microorganisms, colloidal particles), raising questions about the generalizability of the results.
  8. Improvement of limitations and application prospects analysis: The discussion should clearly compare the equipment costs and energy consumption of NB technology with traditional processes (e.g., ultrasound, gas sparging), specify the applicable membrane types and whey concentration ranges, and propose optimization schemes such as “continuous NB generation combined with UF” to inform industrial applications.

This study offers an innovative technological pathway for optimizing membrane filtration processes in the dairy industry, with core conclusions that have academic value and industrial application potential. However, shortcomings in the elucidation of NB mechanisms, standardization of experimental design, depth of data analysis, and validation in practical applications currently limit its scientific rigor and translational value. It is recommended that the authors address the above suggestions to improve the manuscript; if these key issues are resolved, the revised manuscript will significantly enhance its rigor and persuasiveness, providing a stronger theoretical foundation for the application of NB technology in membrane separation processes within the food industry.

Author Response

Reviewer 2

Response

This study investigates the application of nanobubbles (NBs) in the ultrafiltration (UF) process of cheese whey, systematically examining for the first time the effects of NBs on membrane permeate flux, concentration factor, physicochemical properties of whey, and protein structure. The core findings—such as a 34% increase in peak flux, a 40% improvement in concentration factor, and no alteration of whey protein secondary structure—provide a novel technological pathway for optimizing membrane filtration processes in the dairy industry. The research perspective is innovative, filling a gap in the field and overcoming the limitations of conventional membrane fouling control techniques (e.g., ultrasound, gas sparging). This work is the first to apply NB technology to the cheese whey UF process, demonstrating that NBs can significantly enhance filtration efficiency by mitigating concentration polarization and suppressing membrane fouling. The experimental design is systematic and well-supported by data, employing a multidimensional research framework that integrates “process performance, physicochemical properties, and structural characterization.” UF experiments monitored changes in permeate flux and concentration factor, complemented by physicochemical analyses (total solids, protein content), and protein structural stability was validated using FTIR, SDS-PAGE, particle size, and zeta potential measurements. The study is application-oriented with significant industrial value, addressing critical issues such as flux decline and increased energy consumption caused by membrane fouling in dairy UF processes. NB technology can shorten the time required to achieve the target solid content in whey concentration by 30%, offering both energy-saving and high-efficiency advantages. However, the manuscript has several issues that need systematic improvement to enhance its scientific rigor and translational value.

Thank you for reviewing the manuscript and providing valuable insights to improve it.

The mechanism description is overly general: It only proposes two pathways—“electrostatic hindrance” and “turbulence and shear force generated by bubble rupture”—without clarifying the specific interactions between NBs and whey foulants such as proteins and fats (e.g., hydrogen bonding, hydrophobic interactions), making it difficult to explain why NBs can selectively inhibit whey protein deposition on the membrane surface.

Thanks for your comment.

In our study, we didn’t find any specific interactions of NBs with model cheese whey from FTIR and SDS-PAGE analysis.

This would need future work to understand the mechanism precisely at membrane surface. We have included the statement below in the manuscript:

Future studies should focus on understanding the interactions precisely at molecular level between whey proteins and nanobubbles including the interactions at membrane surface during UF. (L 470-472)

Lack of microscopic characterization evidence: There is no microscopic observation of the membrane fouling layer (e.g., SEM, AFM), so the direct impact of NBs on fouling layer thickness and density cannot be confirmed, nor is there a quantitative analysis of the shear force generated by NB rupture.

Thanks for your comment. We agree that microscopic studies would have been valuable, however, we could not perform it in our study.

Although, previous studies have reported and shown the effect of nanobubbles on membrane surface using microscopic characterization.

We have included the information available from previous studies in the mechanism section. (L 456-458)

Missing correlation between NB properties and effects: Key parameters of the generated NBs (e.g., particle size distribution, concentration, zeta potential, stability) are not characterized, making it impossible to establish a quantitative relationship between NB properties and their effects, such as the impact of different NB sizes or concentrations on flux improvement.

The nanobubble properties such as particle size, zeta potential and stability were studied in our other work, which is yet to be published. We optimized the nanobubble generation conditions from our previous studies and has been referred to in the methods section. (L 121-123)

Ambiguity in NB generation process parameters: The manuscript only mentions “oxidizer circulation for 10 minutes, air pressure at 20 PSI, flow rate of 0.25 LPM,” without specifying critical conditions such as liquid temperature and pH during NB generation, nor does it verify the uniform dispersion of NBs in whey after generation.

Thanks for your comment.

We have included these details in the manuscript as suggested. (L 120-122)

The pH of model cheese whey has been mentioned in L 104.

Unclear membrane pretreatment and cleaning procedures: The methods for membrane pretreatment prior to UF experiments (e.g., soaking, cleaning steps), calibration data for initial membrane flux, and post-experiment cleaning and regeneration procedures are not described, affecting experimental repeatability.

We have included these details as suggested in the methods section 2.4. (L 140-144)

Failure to exclude potential confounding factors: The study does not analyze the effects of NBs on the migration behavior of lactose and minerals in whey, nor does it verify whether NBs alter the membrane’s retention characteristics (e.g., differences in retention rates for proteins of varying molecular weights).

Thanks for your comment.

This was not in the scope of our work; however, we have included that this could be a future study to elucidate the migration behavior of different components of milk especially proteins having varying molecular weights. (L 466-472)

Lack of consideration for the complexity of real whey: The experiments use model whey (WPC + lactose + water) with a simple composition, without simulating the complex matrix of actual cheese whey (e.g., free fatty acids, microorganisms, colloidal particles), raising questions about the generalizability of the results.

We have used the model system to be consistent with the stream for all the experiments and avoid day-to-day variation in composition of cheese whey and thereby, affecting the results.

Further, we used WPC 34 as mentioned in the methods which had 8% fat on dry basis. (L 93-94). Therefore, we captured the fat in our model system used in the experiments, hence, we were able to mimic the cheese whey composition.

Improvement of limitations and application prospects analysis: The discussion should clearly compare the equipment costs and energy consumption of NB technology with traditional processes (e.g., ultrasound, gas sparging), specify the applicable membrane types and whey concentration ranges, and propose optimization schemes such as “continuous NB generation combined with UF” to inform industrial applications.

Thank you for this valuable comment.

We acknowledge that a detailed comparison of equipment costs, energy consumption, and application prospects of nanobubble (NB) technology versus traditional processes (e.g., ultrasound, gas sparging) was not within the scope of our current study.

We have mentioned these limitations in the manuscript for future work in section 3.6. (L 465-475)

Our work primarily focused on evaluating the technical feasibility and performance of NBs during ultrafiltration at the laboratory scale. Future studies should focus on understanding the interactions precisely at molecular level between whey proteins and nanobubbles including the interactions at membrane surface during UF. Additionally, there is need for conduct pilot and industrial scale studies to enable commercial scaling up of nanobubbles application in membrane filtration, including techno-economic and energy assessments, compatibility assessment for different membrane materials, pore sizes and streams.

This study offers an innovative technological pathway for optimizing membrane filtration processes in the dairy industry, with core conclusions that have academic value and industrial application potential. However, shortcomings in the elucidation of NB mechanisms, standardization of experimental design, depth of data analysis, and validation in practical applications currently limit its scientific rigor and translational value. It is recommended that the authors address the above suggestions to improve the manuscript; if these key issues are resolved, the revised manuscript will significantly enhance its rigor and persuasiveness, providing a stronger theoretical foundation for the application of NB technology in membrane separation processes within the food industry.

Thank you for your valuable inputs. We really appreciate it.

We have made the suggested changes to improve the manuscript.

Round 2

Reviewer 1 Report

Comments and Suggestions for Authors

The authors have addressed the necessary points in their revision. The changes made are appropriate and improve the manuscript. Therefore, the paper is suitable for acceptance.

Reviewer 2 Report

Comments and Suggestions for Authors

The author has revised the manuscript according to my suggestions, and I believe the changes are very satisfactory. At present, the manuscript can be accepted.